# Engineering domain-inlaid SaCas9 adenine base editors with reduced RNA off-targets and increased on-target DNA editing

Minh Thuan Nguyen Tran 1✉, Mohd Khairul Nizam Mohd Khalid[1], Qi Wang[1], Jacqueline K. R. Walker[1], Grace E. Lidgerwood[2,3], Kimberley L. Dilworth[4], Leszek Lisowski 4,5, Alice Pébay 2,3 & Alex W. Hewitt 1,6

Precision genome engineering has dramatically advanced with the development of CRISPR/ Cas base editing systems that include cytosine base editors and adenine base editors (ABEs). Herein, we compare the editing profile of circularly permuted and domain-inlaid Cas9 base editors, and find that on-target editing is largely maintained following their intradomain insertion, but that structural permutation of the ABE can affect differing RNA off-target events. With this insight, structure-guided design was used to engineer an SaCas9 ABE variant (microABE I744) that has dramatically improved on-target editing efficiency and a reduced RNA-off target footprint compared to current N-terminal linked SaCas9 ABE variants. This represents one of the smallest AAV-deliverable Cas9-ABEs available, which has been optimized for robust on-target activity and RNA-fidelity based upon its stereochemistry.

[1] Menzies Institute for Medical Research, School of Medicine, University of Tasmania, Tasmania, Australia. [2] Department of Surgery, The University of Melbourne, Victoria, Australia. [3] Department of Anatomy and Neuroscience, The University of Melbourne, Victoria, Australia. [4] Translational Vectorology Research Unit, Children's Medical Research Institute, Faculty of Medicine and Health, The University of Sydney, Westmead, Australia. [5] Military Institute of Hygiene and Epidemiology, The Biological Threats Identification and Countermeasure Centre, Puławy, Poland. [6] Centre for Eye Research Australia, The University of Melbourne, Victoria, Australia. ✉email: pherominh1@gmail.com

Cytosine base editors (CBEs) direct cytosine-to-thymine chemistry at a user-defined guide sequence (sgRNA)[1,2], and comprise a cytosine deaminase derived from vertebrate (APOBEC and activation-induced deaminase variants)[3] or invertebrate systems (pmCDA1; Target-AID)[2]. Current generation adenine base editors (ABEs) employ a dimerized, codon optimized variant of laboratory-evolved ecTadA (ABEmax)[4,5], and have directed site-specific adenine-to-guanine nucleotide conversions in a diverse array of systems[6,7]. Despite their broad scope for robust on-target editing, non-engineered ABEs have a significant off-target footprint on the transcriptome and effect incidences of missense and nonsense mutations[8].

Efforts to minimize the occurrence of promiscuous editing have largely improved the fidelity of existing ABEs by installing various inactivating mutations in the wild-type domain of the ecTadA monomer[9], or use truncated variants of ABEmax with amino acid substitution to reduce non-specific contacts with RNA in the recently described, miniABEmax, which consists of a single, evolved ecTadA monomer[10,11]. To date, these strategies are effective at improving the biosafety of ABEs but represent a Cas9-independent solution toward minimizing aberrant editing. Recently, Huang and colleagues described circularly permuted base editors in *Streptococcus pyogenes* Cas9 (SpCas9) and found that they had comparable on-target activity compared to their uncircularized counterparts[12]. Similarly, efforts to engineer a domain-inlaid variant of base editors in SpCas9 have been reported, though this has been less well investigated in terms of their perturbative effects on protein secondary structure and base-editing profile[13].

Here, using insight gained from the profiling of domain-inlaid and circularly permutant SpCas9 base editors, we show that the aberrant RNA off-target effects of ABEs can be modulated based on their overall secondary structure and spatial relation to *Staphylococcus aureus* Cas9 (SaCas9). Our results establish an alternative means for increasing on-target DNA-editing efficiencies while minimizing collateral base editing of RNA transcripts, without introducing amino acid substitutions in the base editor domain. By fine-tuning the spatial positioning between the base editor and Cas9 component, this work represents a useful addendum to efforts enhancing base editing activity and fidelity.

## Results

**Comparison of intradomain and circularly permuted SpCas9-CBE.** The compact size and distinctive cytosine-to-guanine base-editing signature of the hAIDx deaminase (P182X, with 182 residues) made it an ideal candidate for a functional screen of intradomain and circularly permuted base editors (Fig. 1a). First, we selected several sites of interest in the REC2, REC1, and RuvC-III domains of SpCas9 (Supplementary Fig. 1), which were previously shown to be highly amenable to protein-domain insertion without loss of function[14]. Ordinarily, the hAIDx domain is tethered to its C-terminal nickase SpCas9 (nSpCas9) via an N-terminal linker[15]; therefore, we conserved its previously characterized 44-amino acid N-terminal linker and appended a floppy glycine–serine-rich linker to its C-terminus to bridge the nSpCas9 and hAIDx protein domains as a domain-inlaid CBE. To broadly survey the effects of protein-domain alterations on base-editing activity, we also compared three circularly permuted nSpCas9 constructs of interest (Fig. 1b)[16]. Circular permutant variants of the hAIDx base editor at nSpCas9 residues 1010, 1029, and 1058 (Supplementary Fig. 1) were selected for a direct comparison of on-target editing using a cell line expressing yellow fluorescent protein (YFP), which has no homologous analog in the human genome. Collectively, we show that the intradomain insertion of the hAIDx deaminase maintains a consistent on-target DNA

signature (characterized by cytosine-to-guanine transversions at position 9 of the sgRNA) compared to its C-terminal variant, and that nSpCas9 domain-interruptions are most amenable at residue 1058 in our preliminary screen (Fig. 1b; Supplementary Figs. 2 and 3). Comparably, the highest on-target editing was observed with the circular permutation of nSpCas9 at residue 1029 compared to other circular permutant variants of hAIDx (Fig. 1b).

We also found that the intradomain insertion of rAPOBEC1 (BE3) at residue 1058 maintains on-target cytosine-to-thymine activity despite a 2.2-fold average reduction in editing efficiency at the YFP locus (26.9% and 12.1%, respectively). Interestingly, although the C-terminal appendage of a uracil DNA glycosylase inhibitor (UGI) directs product fate toward a cytosine-to-thymine base transition, our head-to-head comparison of CBEs showed that BE3 had the poorest product purity for a construct bearing a UGI (Supplementary Figs. 3 and 4). Altogether these results establish that the inlaying of CBEs at residue 1058 is amenable for the insertion of different varieties of base editors and is sufficiently plastic for dramatic structural variations without deleterious effects (Supplementary Fig. 5).

**Circular permutation and intradomain insertion of SpCas9-ABE variants dramatically affect on-target DNA and off-target RNA editing.** We then generated several conformational variants of ABEmax to profile their on-target DNA and off-target RNA editing efficiencies (Fig. 2a). Initially, we adapted our circularly permuted nSpCas9 designs, which used hAIDx and rAPOBEC1 insertions at position 1029, to an N-terminal, C-terminal, and a decoupled ecTadA dimer variant of ABEmax. On average, the N-terminal circular permutant of ABEmax (ABEmax hereafter referred to as "wildtype") severely impeded editing at the YFP locus compared to its wild-type counterpart (4.2% vs. 40.5%, averaged across three independent technical replicates; Fig. 2b). However, there was a modest, fourfold improvement in editing efficiency at a previously well-characterized locus (ABE site 16; ABE16)[5], compared to the YFP locus. Intriguingly, the C-terminal circular permutant construct had a ten- and fourfold reduction to on-target editing at the YFP and ABE16 loci, respectively, but increased the incidence of localized, RNA off-target events at two promiscuous transcripts (Supplementary Figs. 6 and 7).

Next, we decoupled the ecTadA dimer of ABEmax (Fig. 2a). Although recent literature has suggested that the unevolved ecTadA monomer was dispensable to on-target DNA editing[11,17], we decided to further investigate whether decoupling of the ecTadA monomers influences the on-target editing efficiency of circularly permuted ABEs. Here, we placed the unevolved ecTadA monomer of ABEmax at the C-terminus of the circularly permuted nSpCas9 construct and shifted its evolved monomer to its N-terminus. As expected, decoupling of ABEmax did not significantly affect the on-target activity of the N-terminal circular permutant of ABEmax (Fig. 2b). Surprisingly, however, we found that there was a modest increase in the incidence of localized RNA off-target events at the DNAJB transcript, which was previously observed only by circularly permutating ABEmax at its C-terminus (Fig. 2c, Supplementary Tables 3–6; Supplementary Figs. 6 and 7).

Circularly permuted miniABEmax (V82G) was then investigated. MiniABEmax (V82G) has less RNA off-target activity as it harbors only a single evolved ecTadA monomer and has been engineered for reduced non-specific RNA contact[10]. Circular permutation of the miniABEmax, however, showed no appreciable on-target DNA editing and had no significant bearing on the incidence of off-target RNA events.

Next, we compared the effects of inlaying both the ABEmax and miniABEmax (V82G) variants at our previously characterized

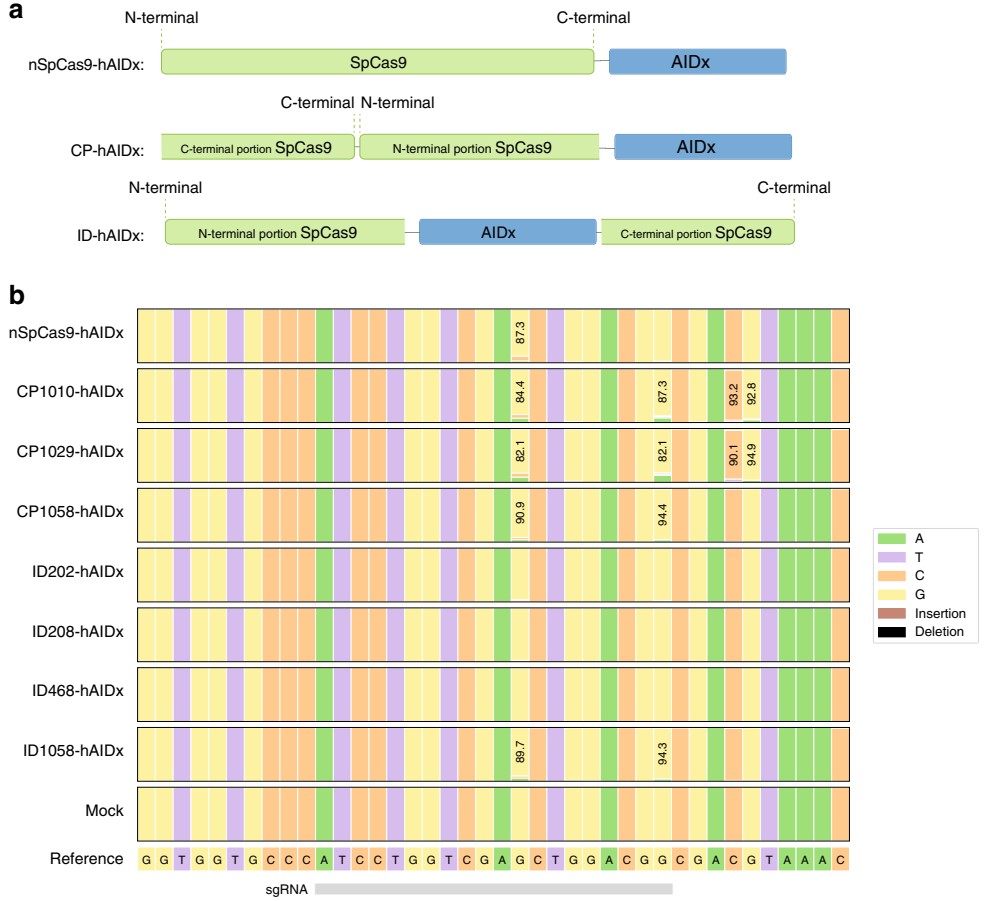

**Fig. 1 Activity profile of nSpCas9-hAIDx permutations. a** Schematic overview of nSpCas9-hAIDx constructs investigated. Here nickase SpCas9 is displayed in green, with cytosine base editors (CBE) shown as a dark blue stadium. **b** Nucleotide quilt of the N-terminal linked hAIDx and circularly permuted (CP) or intradomain (ID) SpCas9 constructs targeting the *YFP* locus. Here, the editing efficiencies and product purities have been averaged across three technical replicates. Source data are available in the Source data file.

intradomain site at residue 1058 in nSpCas9 (Fig. 1a). Overall, we found a 3.5- and 1.7-fold average reduction to on-target editing at the *YFP* and *ABE16* loci, respectively, upon intradomain insertion of ABEmax in nSpCas9 (Supplementary Figs. 8 and 9). Counterintuitively however, we also noted that this permutation resulted in a marked increase in the incidence of RNA off-target events at the *DNAJB* transcript compared to its wild-type counterpart, but reduced off-target events at the *SCAP* transcript (Fig. 2c; Supplementary Figs. 6 and 7). For the miniABEmax (V82G) variant, on-target DNA editing was dramatically abrogated when the evolved ecTadA monomer was inlaid compared to its native conformation, but no appreciable difference to the RNA editing profile was observed (15.5- and 8.5-fold average reductions for *ABE16* and *YFP* loci, respectively). Altogether, these findings suggest that both the DNA and RNA activity profiles of ABEs can be altered based upon their domain positioning in nSpCas9.

**Domain engineering of a minimal ABE fine-tunes base-editing activity based on protein secondary structure**. Given these findings we then designed SaCas9 nickase (nSaCas9)-intradomain ABE constructs. Although the alignment between SaCas9 and SpCas9 crystal structures revealed poor structural homology between the two proteins[18,19], we found that residue 1058 in SpCas9 was conformationally analogous to the poorly crystalized protein loop of residue 745 in SaCas9. Encouraged by these insights, we assayed the length of the uncharacterized protein

loop between residues 730 and 745 within the constraints of the adjacent alpha helices by inserting a base-editing domain at each amino acid position. To further elucidate the apparent positional dependency of base-editing activity and protein structure, we further assayed residues 119 to 132 in nSaCas9 (Supplementary Fig. 10). These residues were positional analogs to the topographically equivalent residue of 468 in nSpCas9, which we assayed in our preliminary screen of intradomain CBE insertions in the REC lobe of nSpCas9 using hAIDx (Supplementary Fig. 1).

We reasoned that the use of the miniABEmax (V82G) variant (Sa-miniABEmax[V82G]) may act as a superior base-editing potentiometer for an activity dependent screen given its comparable on-target efficiency to ABEmax in nSaCas9 (SaABEmax). Interestingly, the insertion of a base-editing domain between residues 119 and 132 in nSaCas9 significantly impeded the on-target activity of the miniABEmax (V82G) (between 0.00 and 5.47% across residues 119–132), whereas on-target activity was dramatically improved when inserted between residues 730 and 745 of nSaCas9 (between 5.39 and 17.7% across residues 730–745). Moreover, a gradated, topographical "hotspot" was revealed by shifting the base editor domain from one residue to another at the assayed positions (Supplementary Fig. 11), until a local "maximum" was achieved with the highest on-target editing efficiency being observed at residue I744 (13.6–17.7% across three independent technical replicates). Here, the insertion of the miniABEmax (V82G) base editor at residue I744 (hereafter referred to as "microABE I744") showed significantly superior

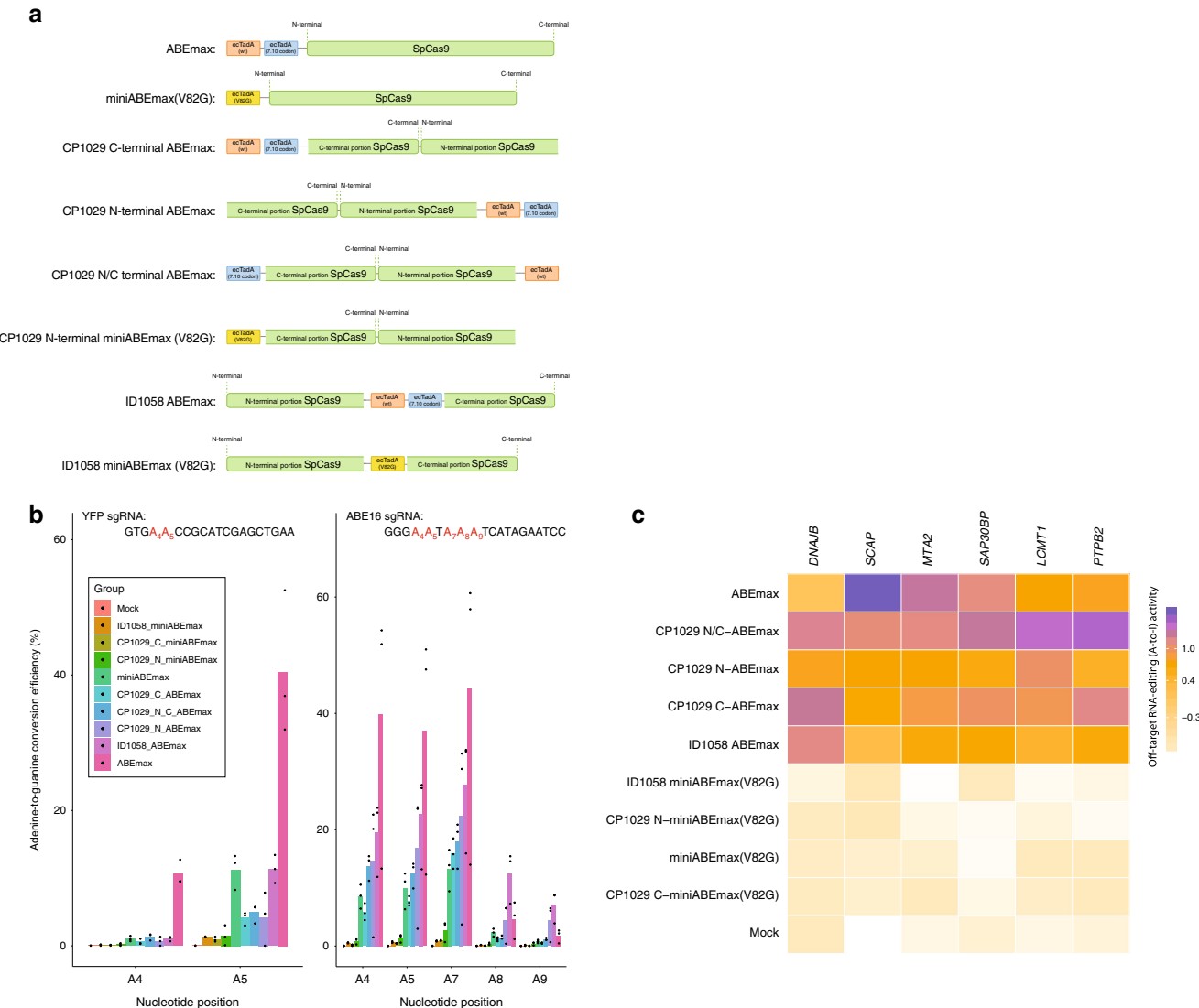

**Fig. 2 Activity profile of nSpCas9-adenine base editor (ABE) permutations. a** The wild-type TadA and the evolved TadA monomer for ABEmax are light blue and orange stadiums, respectively, while the miniABEmaxV82G is depicted as a yellow stadium. **b** On-target activity profile of different ABE permutants tested. nSpCas9-variants of ABE permutants were challenged against the *YFP* and *ABE16* loci. Data are pooled from three independent technical replicates, showing the mean (colored bar) activity at several adenine positions measured for adenine-to-guanine editing. **c** Heatmap showing the off-target profile of different permutations of nSpCas9 ABEs. Off-target events were considered at the position of the most heavily deaminated adenine in the promiscuous *DNAJB, SCAP, MTA2, SAP30BP, LCMT1,* and *PTPB2* transcripts (three independent technical replicates of three biological replicates targeting *YFP, ABE16,* and non-targeting were included). Adenosine (A) to inosine (I) editing efficiencies are shown as the mean value scaled from the lowest to highest value across all samples for all RNA off-target sites. Source data are available in the Source data file.

on-target activity at the *ABE16* locus compared to Sa-miniABEmax (V82G) and SaABEmax (15.96% vs. 1.19% and 1.03%, respectively; Supplementary Fig. 12). Interestingly, the insertion of the hAIDx domain in nSaCas9 was also consistent with the higher on-target editing efficiencies found for the ABEs for position I744 (microAIDx I744), as compared to G129 and N730 (Supplementary Fig. 13). However, the insertion of the hAIDx deaminase at position G129 did not abrogate on-target editing like it did for the insertion of the miniABEmax (V82G) domain at this position. Moreover, we noted that on-target cytosine-to-thymine editing was modestly maintained, albeit with a slightly altered activity window.

**Intradomain insertion can enhance on-target DNA editing and broaden the activity window breadth.** Encouraged by these preliminary results, we then characterized the activity window of

the microABE I744 against 15 previously validated sites. The microABE I744 had a broader activity window with improved, overall on-target editing efficiencies compared to its SaABEmax and Sa-miniABEmax (V82G) counterparts (Fig. 3a, b). We observed up to a 2.28- and 1.78-fold increase in editing efficiency at the A7 position compared to SaABEmax and Sa-miniABEmax (V82G), respectively. At the A10 position, microABE I744 out-performed the SaABEmax and Sa-miniABEmax (V82G) by up to 3.63- and 3.09-fold, respectively. Overall, the microABE I744 vastly augments the editing scope of targettable adenines within a 21-nucleotide spacer, displaying a characteristic bi-lobed activity window spanning from adenine position 4 to 16 (Fig. 3b).

**Intradomain insertion can attenuate the incidence of aberrant off-target RNA editing.** We sought to then characterize the effects of nSaCas9 intradomain base editor insertion on RNA

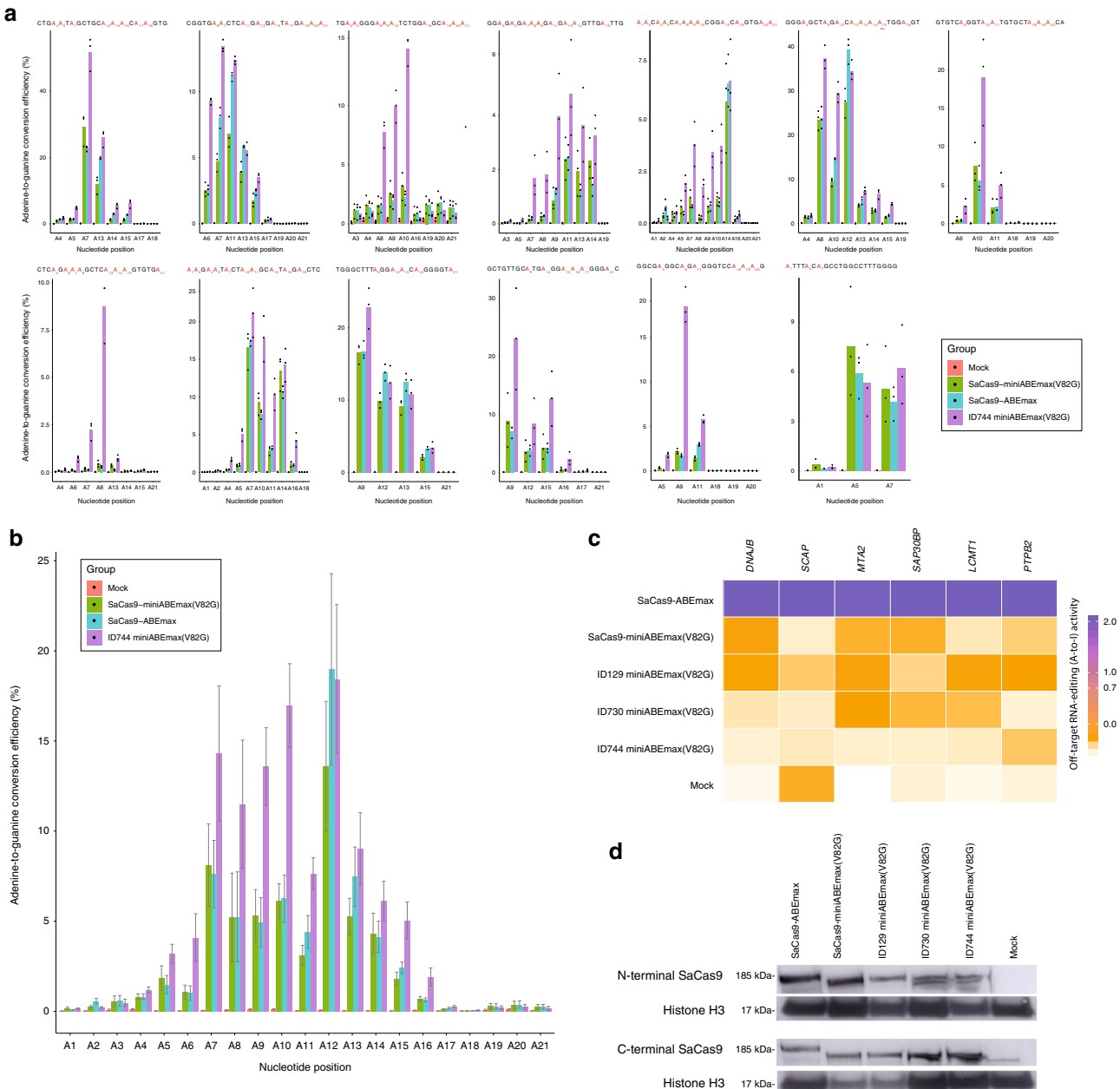

**Fig. 3 Profiling intradomain SaCas9 constructs in HEK293A-YFP cells. a** On-target activity window in HEK293A-YFP cells. Editing efficiency was determined at various adenine positions in a 21-nucleotide activity window using sgRNAs targeting 16 different loci ($n = 3$ independent technical replicates performed on separate days). **b** Merged data of the average base editing showing A-to-G modifications from (**a**) with microABE I744, SaABEmax, and Sa-miniABEmax (V82G), with each corresponding sgRNA. Data are presented as mean $+/-$ SEM derived from a composite of independent sgRNA replicates and their respective technical replicates ($n = 3$ for each sgRNA). **c** Off-target profile of nSaCas9 constructs. Promiscuous RNA transcripts were assayed for the incidence of adenosine-to-inosine editing in the presence of nSaCas9-miniABEmax (V82G), nSaCas9-ABEmax, Intradomain-nSaCas9 I744 (microABE), Intradomain-nSaCas9 N730, and Intradomain-nSaCas9 G129. The most heavily edited adenine nucleotide of each transcript was considered for comparison. MiSeq data was considered across biological and technical replicates ($n = 9$ in total). Adenine (A) to inosine (I) editing efficiencies are shown as the mean value scaled from the lowest to highest value across all samples for all RNA off-target sites. **d** Western blot demonstrating immuno-binding of antibodies targeting the N- and C-terminal of SaCas9 constructs for each base editor tested ($n = 3$ western blots performed on independent days). Source data are available in the Source data file.

activity. Here, we reasoned that the inlaying of a base editor domain could further attenuate the incidence of RNA off-target events by exerting either a steric limitation on the deaminase domain, or by altering the secondary structural folding and expression of the base editor. In addition to assaying the micro-ABE I744, the intradomain insertion of miniABEmax (V82G) base editors at residues G129 (Sa-ID129 miniABEmax (V82G)) and N730 (Sa-ID730 miniABEmax (V82G)) were also challenged

against an adenine-rich *RNF2* locus, a previously validated sgRNA against *ABE16*, and a non-targeting sgRNA against *LacZ*. Here, the microABE I744 reduced the breadth of off-target events for at least three of the six commonly deaminated RNA off-target transcripts compared to the Sa-miniABEmax (V82G) and SaA-BEmax (Fig. 3c; Supplementary Table 3). Strikingly, the micro-ABE I744 also had a significantly reduced, local RNA off-target profile compared to its counterparts at residues G129 and N730

by up to 2- and 1.8-fold, respectively. Interestingly however, ABE insertion at residue G129 dramatically increased the incidence of RNA off-target events even relative to the Sa-miniABEmax (V82G) (Fig. 3c; Supplementary Figs. 14 and 15). Next, we wanted to determine if whether these differences in RNA off-target effects were due to variations in protein expression or protein folding of the domain-inlaid base editors[10]. We performed western blots with primary antibodies targeting the N-terminus of domain-inlaid nSaCas9 base editors and the C-terminal flag tag of the respective constructs. Overall, there was no major difference in the expected banding patterns for each construct. Taken together, these results indicate that it was unlikely that RNA off-target-specific differences were attributable to protein folding specific variations, such as premature stop codons occurring within the open reading frame (Fig. 3d; Supplementary Fig. 16).

Finally, RNA-seq was used to characterize the molecular footprint of the microABE I744, SaABEmax, and Sa-miniABEmax (V82G) on the transcriptome. The microABE I744 dramatically lowered the incidence of aberrant mRNA off-target events compared to both the SaABEmax and Sa-miniABEmax (V82G) (2243 reads containing adenosine-to-inosine editing for microABE I744 as compared to 4425 and 52,030 reads for Sa-miniABEmax [V82G] and SaABEmax, respectively). In some instances, the domain-inlaid base editor resulted in a sixfold reduction in the number of mRNA off-target edits (Supplementary Fig. 17) as compared to its non-inlaid permutant (Sa-miniABEmax (V82G)) (81 vs. 544 reads containing A-to-I editing, respectively, for transcripts mapped to chromosome 19). Here, we postulate that the altered positioning of the deaminating catalytic pocket is "hidden" from the circulating RNA transcripts, though we cannot definitively preclude other mechanisms that would affect the RNA mutagenicity of domain-inlaid ABEs without crystallographic structures.

To assay whether domain-inlaid ABEs adversely affected their DNA-editing fidelity, we selected the top 28 predicted gDNA off-target sites based on the sgRNA-target homology[20] of the top three edited sites (ABE site 11, *ABE11*; ABE site 8, *ABE8*; ABE site 1, *ABE1*). We found that, overall, there was no apparent change in the off-target DNA-editing breadth of the microABE I744 as compared to its existing counterparts at putative off-target sites (Supplementary Data 1). Whole-exome sequencing was further performed at a depth of 1000× for a less biased measure of DNA-editing fidelity. In support with previous results, we found that the off-target DNA fidelity of the microABE I744 did not change relative to SaABEmax or Sa-miniABEmax (V82G) (between 13 and 26 A-to-G conversions relative to normalized control samples; $n = 3$).

**Domain-inlaid ABEs enables correction of disease-specific loci and single-vector AAV-mediated delivery**. We then directed the microABE I744 to correct the highly penetrant *PCDH15* Arg245Ter variant, which causes type 1 Usher syndrome, whereby homozygous carriers have congenital deafness and develop retinitis pigmentosa[21]. We observed a 10-fold increase in editing efficiency and dramatically lower mRNA off-target effects as compared to Sa-miniABEmax (V82G) ("Methods"; Supplementary Table 5). As expected, on-target editing was abrogated upon introduction of the SaKKH-related mutations, which imposes an incompatible preference for a canonical thymine at position 6 at the NNGRRN PAM of our sgRNA targeting the *PCDH15* Arg245Ter variant (Supplementary Figs. 18 and 19).

Finally, we sought to demonstrate that the microABE I744 can be packaged as an all-in-one vector for adeno-associated viral

(AAV) delivery[22]. Current generation AAV-mediated delivery platforms for base editors employ a dual-vector system, which is largely reliant on the use of intein *trans*-splicing for the reconstitution of full-length CBE or ABE[23]. This can hamper on-target editing efficiencies due to the need for co-delivery and co-transduction of the payload. As proof-in-principle, we targeted the previously well-characterized locus, *ABE11*, and show that our all-in-one vector can be packaged as AAV-7m8 and AAV-DJ serotypes. To fit within the packaging constraints of the AAV vector, we package the minimal SCP1 promoter to drive microABE I744 expression[24], a single mammalian terminator (bgH polyA), and a hU6 promoter with sgRNA targeting *ABE11* or non-targeting *LacZ*. Next, we adapted the single-stranded DNA virus sequencing platform and show that no apparent truncation of the virus has occurred at either the 5′ or 3′ inverted tandem repeats (ITRs), and that genomic rearrangement events were few (Fig. 4a, b)[25]. With this insight, we transduced HEK293A-YFP cells and observed an editing efficiency of ~0.24% with no selection or enrichment after only three days of culturing with either the 7m8 and DJ capsid derivatives (Fig. 4c). Similarly, when the 7m8 and DJ AAV-serotypes were applied to terminally differentiated iPSC-derived retinal optic cups at a modest viral titer, we found that the AAV-7m8 serotype induced editing of the organoids after only 7 days of non-selective culturing (Supplementary Fig. 20).

## Discussion

Overall, the activity profile of the ABEs can be improved for their on-target efficiency and precision by manipulating the structure of Cas9. Although previous research has characterized the effects of protein engineering on the ABE[9,10,12,26], our work further expands upon these efforts by refining the spatial arrangement between the endonuclease and base editor components. We show that the same variant of ABE can have different DNA and RNA editing profiles arising from alterations to their secondary structure. Through the strategic use of circular permutation and protein-domain insertion, we observe that both the DNA and RNA footprint can be calibrated based upon a model of "best-fit."

We found that the adaptation of ABEmax in nSaCas9 had significantly lower on-target editing activity compared to its nSpCas9 counterpart, possibly due to protein-specific differences between SaCas9 and SpCas9[17]. Likewise, the use of the recently described miniABEmax variant in nSpCas9 showed robust on-target editing in its native, N-terminal conformation, but failed to show appreciable editing in the same permutation in nSaCas9. Interestingly, however, on-target efficiency was entirely abrogated when miniABEmax was inserted as an intradomain construct in nSpCas9, but conversely was enhanced upon its insertion at its positional analog in nSaCas9. Although the development of the miniABEmax suggests that off-target activity can be inherently minimized as a Cas9-independent solution through amino acid substitutions and deletions, we show that these effects appear to be particular to a specific, overall secondary structure.

In contrast to the vastly superior on-target and reduced off-target capabilities of the microABE I744, the intradomain miniABEmax variant at residue G129 showed a counterintuitive increase in the incidence of RNA off-target events and an overall reduction to on-target editing efficiency. These effects were not due to obvious differences in protein expression or protein folding (Fig. 3d). Taken together, we believe that inlaid base editors may sterically hinder the stochastic movement of the deaminase domain from freely circulating RNA transcripts, though it is impossible to make such an assertion without further crystallographic structures.

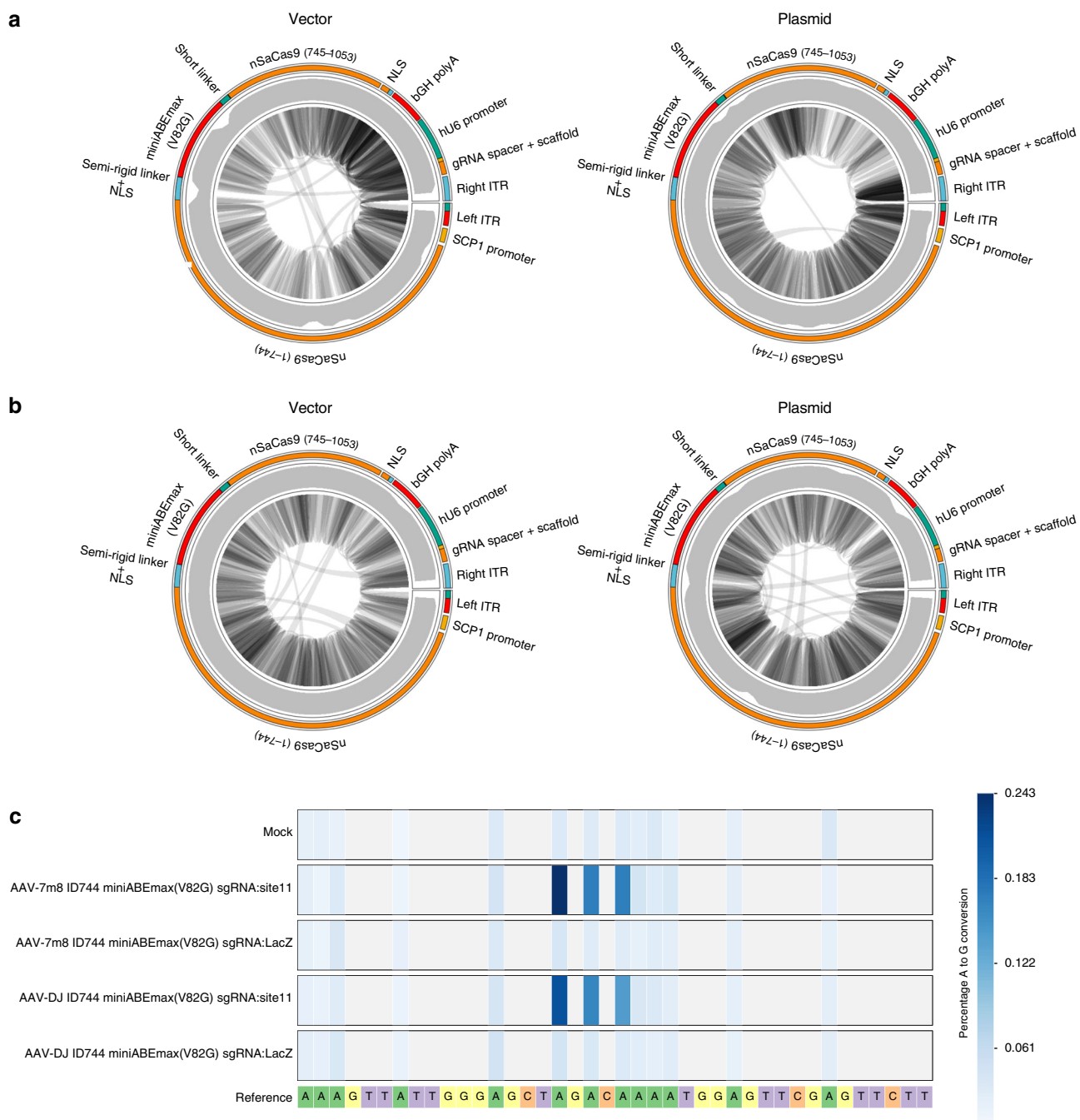

**Fig. 4 AAV-mediated delivery of a single construct containing domain-inlaid nSaCas9 I744 (microABE I744) and a sgRNA targeting either ABE site 11 or LacZ. a** Schematic showing the all-in-one AAV construct driving the expression of the microABE I744 via an SCP1 minimal promoter and a hU6 promoter for the sgRNA component, with example aligned reads from 7m8-AAV vector and its corresponding plasmid, as well as (**b**) AAV-DJ packaged vector and its corresponding plasmid. **c** Percentage of A-to-G conversions are presented as an average of three technical replicates, for either the AAV-7m8 or AAV-DJ serotypes, in HEK293A-YFP cells. Source data are available in the Source data file.

When the microABE I744 was packaged into an AAV-deliverable format, we found somewhat modest editing in both dividing and non-dividing cell types. Deep sequencing of the AAV-packaged viral genomes revealed that editing efficiency was not affected by truncations at the ITRs or due to genomic rearrangements. Currently, it is unlikely that the editing efficiency of our single AAV vector-packaged microABE I744 has surpassed a therapeutic threshold. Nonetheless, given our promising in vitro plasmid-based results, future directions could consider further

optimization to the AAV payload architecture by placing the U6-sgRNA component in the antisense direction, or adding additional regulatory elements for enhanced protein expression, as well as comparing our single-vector format against dual-vector constructs such as packaged SaABEmax or through the screening of different promoter sequences[23].

In summary, we show that the manipulation of the Cas9 secondary structure can further augment the precision of ABEs by carefully considering the broader, steric relationship

between Cas9 and base editors. At only 3.8 kb in size, it is small enough to fit within the constraints of an AAV vector with adequate packaging space for a promoter and its cognate guide sequence. In addition, the broad editing window of the microABE I744 and its robust on-target editing and reduced RNA signature on the transcriptome makes it an ideal candidate for further preclinical testing.

## Methods

**PyMOL analysis and I-TASSER alignment of SpCas9 and SaCas9.** Crystal structures of *S. pyogenes* Cas9 (PDB accession 4OO8) and *S. aureus* Cas9 (PDB accession 5CZZ) were downloaded from the Protein Data Bank and visualized using PyMOL v2.3.1[27]. Given that residues 731–741 in SaCas9 were not crystalised, I-TASSER was used to generate a predictive crystal structure (available upon request) that was then superimposed with that of *S. pyogenes* Cas9 using the "super" command in PyMOL[28]. The "complete" SaCas9 structure was then aligned structurally using the TM-Align webtool from I-TASSER to determine the structural homology between the two proteins[28].

**Plasmid construction and cloning.** Plasmids were generated and Sanger sequence verified by Genscript (Piscataway) (Supplementary Table 1). Plasmids expressing the U6-sgRNA scaffold with mCherry fluorophore reporter were cloned into either the pX552-CMV-mCherry-U6-SpCas9_sgRNA scaffold (Addgene #107051) or PX552-CMV-mCherry-U6-SaCas9_sgRNA scaffold (Addgene #107053) via SapI (NEB) digest sites using oligonucleotides corresponding to the target spacer (Supplementary Table 2; Supplementary Data 2).

**AAV packaging and single-stranded virus sequencing.** The AAV constructs were packaged into Recombinant recombinant AAV (rAAV) vectors particles were produced using a standard transient transfection HEK293 cells[29]. Briefly, HEK293 cells were triple transfected using PEI (Polysciences Cat#239662) with pAd5 helper plasmid[29,30], pAAV transfer vector and AAV-helper plasmid encoding rep2 and the capsid of interest (packaging using pX551, and pseudo-serotyped with the DJ or 7m8) capsid. Viral Packaged vector particles vectors were purified using iodixanol-based density gradients[31], and vector genomes were titred by real-time quantitative PCR (RT-qPCR) as previously described[32]. Single-stranded virus sequencing is described by Lecomte and colleagues[25]. Briefly, each AAV vector was treated with DNaseI, Proteinase K, and RNase A. An internal normalizer was assembled containing each of the DNA species that could be found in the AAV preparation; adenovirus helper plasmid, rep-cap helper plasmid, the transfer plasmid containing the vector genome, the transfer plasmid backbone and HEK293T genomic DNA. All samples then underwent a DNA clean-up step. Second strand synthesis was performed by hybridizing a random hexanucleotide mix (random primer 6, cat#1230S NEB) using DNA pol I (cat#M0209S NEB). Samples then underwent an additional DNA clean-up step followed by library prep and Illumina MiSeq sequencing.

**Cell culture.** HEK293A cells (R70507, ThermoFisher Scientific) expressing yellow fluorescent protein (HEK293A-YFP), which we previously generated[33], were cultured in Dulbecco's modified Eagle medium (DMEM) with high glucose (Life Technologies). Culture media was supplemented with 10% (vol/vol) Fetal Bovine Serum (Life Technologies) and 1% (vol/vol) antibiotic-antimycotic (Thermofisher Scientific). HEK293A-YFP cells were maintained in the aforementioned media at 37 °C with 5% $CO_2$ for cell culture experiments. Cell culture work used HEK293A-YFP cells that were less than 20 passages old. Cells carrying the full *PCDH15* cDNA sequence with the Arg245ter (NM_033056.4:c.733C>T) variant were generated using the Flp-In T-Rex core kit on a Flp-In T-Rex cell background (R78007, Invitrogen, ThermoFisher Scientific) as per manufacturer's instructions and maintained similar to HEK293A lines. H9 human embryonic stem cells (WA09; WiCell) were differentiated into retinal cup organoids using the protocol described by Reichman and colleagues[34]. After terminal differentiation for 30 weeks, optic cups were chosen for the final AAV experiments. Mycoplasma testing was performed on a biweekly basis using PCR Mycoplasma Test Kit I/C (Banksia Scientific).

**Transfections and DNA/RNA extractions.** For on-target DNA and off-target RNA characterization, HEK293A-YFP cells were seeded at a density of 50,000 cells per well in a 24-well, tissue culture-treated plate (In Vitro Technologies). Subsequently, 8 μL ViaFect Transfection reagent (Promega) with 1 μg CRISPR base editor plasmid and 1 μg sgRNA-expressing plasmid was transfected into cells 20–24 h after plating. Fresh media containing 20 μg/mL Blasticidine (Sigma-Aldrich) was exchanged 18–22 h after transfection to select for cells expressing the base editor construct. Further enrichment was performed 18–22 h following the first selection round with the replacement of media containing 30 μg/mL Blasticidine. Overall, cells were cultured for strictly no longer than 72 h after initial transfection before washing with ×1 PBS (ThermFisher Scientific) due to the loss of RNA A-to-I edits in the transcriptome over time. For the initial on-target gDNA editing screen of nSaCas9 intradomain constructs however, total culturing time was

5 days to ensure for maximum selection with Blasticidine, and were extracted for DNA only. For those experiments involving 3 days of culturing, RNA and DNA were simultaneously harvested using 350 μL Buffer RLT Plus as part of the Allprep DNA/RNA Mini Kit (QIAGEN) following the manufacturer's protocol. *PCDH15* Arg245Ter Flp-In T-Rex lines were transfected with 1 μg base editor construct and 0.45 μg sgRNA plasmid (FugeneHD™, Promega), and selected with 1 μg/mL puromycin for 5 days. DNA and RNA samples were eluted in 30 μL Buffer EB and RNase-free water, respectively, with 1.5 μL RNaseOUT Recombinant Ribonuclease Inhibitor (Life Technologies) added to the eluted RNA sample. For experiments involving AAV-transduction, HEK293A-YFP cells were plated at a density of 50,000 cells per well, 24 h prior to transduction at a multiplicity-of-infection (MOI) of $2 \times 10^6$ viral genomes/cell. After 72 h of culture, cells were washed twice with PBS and harvested. Retinal organoids were transduced with $8.0 \times 10^{10}$ to $1.2 \times 10^{11}$ viral genomes for 7 days without selection and harvested.

**Western blot of domain-inlaid base editors.** In a 6-well plate, 200,000 HEK293A-YFP cells were plated 1 day prior and transfected with 2.5 μg of plasmid DNA expressing domain-inlaid base editors in triplicates as detailed above. Cells were harvested according to manufacturer's instructions using RIPA Lysis and Extraction Buffer (Life Technologies) and Halt™ Protease Inhibitor Cocktail (1X) (Life Technologies) after 72 h of culturing as detailed above. Lysate concentrations were normalized using the Pierce™ BCA Protein Assay Kit (Life Technologies) according to manufacturer's instructions, and 40 μg of reduced protein was loaded into each gel (Bolt™Mini Gels; Life Technologies) and ran for 1 h at 130 V. Transfer was performed using the iBlot™ 2 System (Life Technologies) using the following settings: 20 V for 1 min, 23 V for 8 min, 25 V for 4 min. Blocking was performed at room temperature for 1 h with blocking buffer: 5% skim milk (Woolworth, #2885) in 1X TBST (20 mM Tris, 150 mM NaCl, 0.5% Tween 20, pH 7.6). All subsequent washes were performed in triplicates using 1X TBST for 5 min at a time. Membranes were then incubated in primary antibodies diluted in 1X TBST at 4 °C with gentle agitation overnight. For western blot experiments involving HRP-conjugated primary antibodies against the N-terminus of domain-inlaid base editors, a 1:4000 dilution ratio was used (*S. aureus* CRISPR/Cas9 antibody; C15200230-100, Custom Sciences). For those experiments involving the C-terminus of domain-inlaid base editors, a 1:750 dilution ratio was used (DYKDDDDK Tag monoclonal antibody MA1-91878, Life Technologies). Histone H3 was used as a loading control for all experiments and diluted in a 1:4000 ratio (H3pan Antibody 1B1B2, C15200011, Custom Sciences). Following a washing step, the membranes were incubated in a secondary antibody diluted in a 1:4000 ratio in 1X TBST (goat anti-mouse IgG (H + L) secondary antibody, HRP, 31430, Life Technologies) for 1 h at room temperature with gentle agitation. Membranes were washed again and incubated with chemiluminescence buffer (SuperSignal™ West Pico PLUS Chemiluminescent Substrate, 34577, Life Technologies) on a transparent film according to manufacturer's instructions and imaged using the Amersham™ Imager 600. Densitometry analysis was performed using the in-built function with default and "high sensitivity" settings to derive chemiluminescent intensity of the protein bands. Relative chemiluminescent intensity to the loading control was calculated by dividing the intensity for the protein band-of-interest by the signal for the loading control for each well on the same image.

**RNA reverse transcription and targeted PCR amplification.** Between 200 and 400 ng of RNA was reverse transcribed using the High-Capacity RNA-to-cDNA™ Kit (Life Technologies) following the manufacturer's instructions. RNA samples were paired with their counterpart gDNA samples for targeted amplification. The cDNA samples were diluted 1:10 and 2 μL of the diluted cDNA was used as input for the first-round PCR amplification of either RNA off-target sites or undiluted gDNA for those experiments involving DNA on-target sites (Supplementary Table 2). Briefly, PCR reactions were made up to 25 μL comprising 12.5 μL Q5 Hot Start High-Fidelity 2X Master Mix (NEB), 1.25 μL of forward and reverse primers containing 5′ flanking illumina style adapter overhangs, and diluted cDNA or 50–100 ng of gDNA under thermocycling conditions of 98 °C initial denaturation for 30 s, and 30 cycles of 98 °C denaturation for 10 s, 65 °C annealing for 30 s, and 72 °C extension for 12 s with a 72 °C final extension for 2 m. PCR amplification was validated using electrophoresis using 1.5% agarose gel and cleaned using Agencourt AMPure XP (Beckman Coulter) 1.8X paramagnetic bead cleanup.

**RNA-seq analysis.** Sequencing libraries were prepared using NEBNext(R) UltraTM RNA Library Prep Kit for Illumina(R) and sequencing was carried out on HiSeq X Ten using a 2×150-bp paired-end configuration at Genewiz (Suzhou, China). Libraries were downsampled to 120 million reads using seqtk v.1.3 (r106) (https://github.com/lh3/seqtk). The downsampled libraries were processed according to GATK best practices for RNA-seq variant calling[10]. Briefly, raw sequencing reads were aligned to the human hg38 reference genome using STAR (v.2.7.2b). Next, tools from GATK (v.4.1.3.0) that include MarkDuplicates, SplitNCigarReads, BaseRecalibrator, and ApplyBQSR were used to process the aligned reads. Known variants in dbSNP build 138 were used for base quality recalibration. Finally, "analysis-ready" BAM files were subjected to bam-readcount and HaplotypeCaller to estimate per-library nucleotide abundances per position and to identify RNA base-editing variants, respectively. Total A-to-I edits per

library were calculated as the sum of A-I edits on the positive strand and T-C edits on the negative strand.

**Whole-exome sequencing analysis.** Whole-exome sequencing was performed on MGI DNBSEQ-G400 using a 2×150-bp paired-end configuration at Genewiz (Suzhou, China). A workflow similar to the RNA-seq analysis was used. In brief, libraries were downsampled to 510 million reads using seqtk v.1.3 (r106) (https://github.com/lh3/seqtk) and were processed according to GATK best practices. Tools from GATK (v.4.1.3.0) were used for paired-end alignment, removal of duplicated reads and base quality recalibration. The "analysis-ready" BAM files were subjected to the same filtering pipeline as described in RNA-seq analysis. DNA-editing rates attributed to the base editors were calculated by subtracting the background rates of A-to-G and T-to-C substitutions in the control sample from the base editor-treated sample.

**Library preparation for targeted amplicon sequencing.** Following the first-round PCR amplification and cleanup of amplicons containing on-target sites or RNA off-target sites, a second-round barcoding PCR was performed using between 20 and 150 ng of the purified first-round PCR products. The barcoding PCR added unique dual i5/i7 indices using the Nextera XT index kit V2 (Illumina). Q5 Hot Start High-Fidelity 2X Master Mix was used following manufacturer's instructions for a total volume of 25 µL, with 2.5 µL of i5 and i7 Nextera XT indices added, followed by thermocycling conditions as described: 95 °C for 2 m, then 15 cycles of (95 °C for 15 s, 61 °C for 20 s, and 72 °C for 20 s), followed by a final 72 °C extension of 2 m[5]. Subsequently, the second-round PCR products were purified using 0.7× paramagnetic bead cleanup and quantified using Qubit™ dsDNA BR Assay Kit (Life Technologies). Each sample was then normalized to 4 nM and 5 µL of each library member was pooled into a final library that was validated using High Sensitivity D1000 ScreenTape (Agilent Technologies). The final library was paired-end sequenced (2 × 251) on the Illumina MiSeq machine using 600-cycle MiSeq Reagent Kit v3.

**Amplicon sequencing analysis.** Paired-end fastq files were joined and trimmed[35], before being processed using the CRISPResso2 (V.2.0.29) workflow[36]. For the specific calculation of off-target RNA A-to-I editing, amplicons were PCR amplified following reverse transcription and cDNA synthesis as described above, using the primer set for DNAJB1, MTA2, PTBP2, SAP30BP, LCMT1, and SCAP. In addition to comparing the editing frequency of the most highly edited adenine nucleotide position in each amplicon, we summed all A-to-I nucleotide conversions across all relevant sites of each individual amplification. Heatmaps quantifying the off-target profiles were generated in R (v. 1.2.5019) using the "superheat" package.

**Statistical analysis.** The average nucleotide modification percentage outputs from CRISPResso2 (V.2.0.29) were pooled across independent biological and technical replicates for each nucleotide position in the amplicon. Welch Two-sample $t$-tests were performed to compare differences in editing efficiencies, and a $p$ value of <0.005 was considered statistically significant. As outlined above, specifically, for comparative analyses of the six RNA off-target transcripts (*DNAJB*, *MTA2*, *PTBP2*, *SAP30BP*, *LCMT1*, and *SCAP*), both average adenine-to-guanine (inosine) editing across the length of the amplicon, and also the highest edited position of the amplicon were considered[10]. The average of the amplicon was considered as multiple off-target events were observed relative to the untransfected mock control.

**Reporting summary.** Further information on research design is available in the Nature Research Reporting Summary linked to this article.

## Data availability
All raw sequencing reads have been uploaded to the European Nucleotide Archive under the accessions: PRJEB35675 (MiSeq sequencing); PRJEB38819 (RNA-seq profiling); and PRJEB38622 (whole-exome sequencing).

Previously determined crystal structures for SpCas9 and SaCas9 are available from the Protein Data Bank at https://www.rcsb.org/structure/4oo8 and https://www.rcsb.org/structure/5CZZ, respectively. Any other relevant data are available from the authors upon reasonable request. Source data are provided with this paper.

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

## Acknowledgements

This work was supported by a National Health and Medical Research Council (NHMRC) Senior Research Fellowship (A.P., 1154389) and a Practitioner Fellowship (A.W.H., APP1103329), the Australian Research Council Special Research Initiative in Stem Cell Science (Stem Cells Australia), NHMRC project grant, Vector and Genome Engineering Facility, and the Australian Medical Research Future Fund. We are grateful for scripts used in the identification of off-target base editing in RNA-seq data provided by Sowmya Iyer. We thank G.S. Liu, A.L. Cook, K. Fairfax, and F. Patterson for their assistance with cell culture and DNA extraction. In addition, we thank R. KC for his assistance with western blots and J. Marthick for the maintenance of the Illumina MiSeq machine.

## Author contributions

M.T.N.T. and A.W.H. conceived this work. M.T.N.T, M.K.N.M.K., Q.W., J.K.R.W., G.E.L., K.L.D., and L.L. generated reagents and conducted experiments. M.T.N.T., M.K.N.M.K., and A.W.H. performed computational analyses. A.P. and A.W.H. supervised the research. All authors contributed to writing the manuscript.

## Competing interests

The authors declare the following competing interest: M.T.N.T. and A.W.H. have filed a provisional patent application (Australian Provisional Application No. 2020900913) on intradomain SaCas9 base editors. All other authors declare no competing interests.
