## [Peer Review File · Nature Communications]

Reviewers' Comments:

Reviewer #1:

Remarks to the Author:

In this article, the authors engineer a set of base editing proteins by fusing CBE/ABE domains to circularly permuted (CP) or inserting CBE/ABE domains (ID) inside SpCas9/SaCas9. The on-target and off-target editing profiles of these Cas9 base editors are evaluated using DNA and RNA sequencing analysis. The authors found that intradomain insertion of ABE in SaCas9 at position I744 improved the DNA on-target editing efficiency and reduced RNA off-target editing compared to N-terminal fusion of ABE to SaCas9.

In general, the authors provide an interesting method to increase on-target but reduce promiscuous RNA off-target base editing efficiency. However, the manuscript is not well written and the statements are not convincing with current data. The annotation of engineered Cas9 is confusing and not consistent in the main text, figure legends and tables. A concise nomenclature of Cas9 derivatives is suggested. Many labels used in the figures are not explained in the figure legend and the main text. The quality of figures need to be improved to make results more explicit to readers.

Comments:

1. The schematic in Fig. 1a lacks of detailed information on how each construct is made. The DNA construct map (similar to Fig. 3a) should be provided. The nomenclature of the construct is confusing and inconsistent.
2. The editing window for different engineered SpCas9 may be different. In Fig. 1b, only cytosine-to-guanine transversion efficiency at position 9 of the sgRNA is presented. The editing efficiency of other positions are also needed to compare the editing efficiency. The Fig. 1b is suggested to move to the supplementary file as this result is not related to the core topic (Cas9 ABE). The results of this figure are not clearly described in the main text on Page 5.
3. In Fig. 1c, the nomenclature is quite confusing and it is hard to read.
4. In Fig. 1d, 2c and Supplementary Fig. 6, the calculation of the off-target RNA editing (A-to-I) activity should be provided in figure legend.
5. Fig. 2a is suggested to move to Supplementary Information.
6. In Fig. 2b, how to calculate the composite A-to-G efficiency should be described in the figure legend. Meanwhile, the data for each sgRNA targeting locus should be included in the figure or in Supplementary Information.
7. In Fig. 2d, the distribution of off-target editing in chromosomes seems unnecessary information. The total number of off-target editing occurrences should be provided.
8. In Fig. 3, the positive control (e.g. SaCas9-ABEmax in one or two AAV vector(s)) is missing. The scale of proportion of A to G conversion is different from what are used in other figures, which makes the results hard to compare with each other. The description of semi-rigid linker is missing.
9. In Supplementary Fig. 5, no statistical data are provided. Therefore, the "significantly" used in the statement on page 5 is inappropriate.

Reviewer #2:

Remarks to the Author:

In this paper Tran et al. design and test a large number of interesting base editor fusions in HEK293 cells based on extrapolations from extensive prior work done in the CRISPR field. Nevertheless their construct design is astute and by combining and directly comparing a number of past observations they validate much of this previous work and help to bolster previous hypotheses concerning the modular and conserved nature of domain insertion. Moreover they design a specific ABE construct that may be of use in AAV based delivery in the future.

I find this work to be both useful and of high interest and recommend publication with few

modifications.

Major Issues:

In order to better demonstrate the importance of this work please include an experiment demonstrating the activity in at least one, preferably multiple, cell types other than HEK293s. Preferably, a primary cell type. Any animal model, reporter or otherwise with AAV would be ideal but animal demonstration should not be required for publishing

A number of times throughout the paper comments are made about "stereochemistry" dictating the molecular activity of fusions and intercolations. The use of the term stereochemistry is not only confusing, it is misleading as most of the molecules tested are not the same in composition and therefore can not be construed to be stereoisomers. Please rephrase.

The assertion that the difference of effects observed is due solely to the steric limitations of specific fusions, while likely, is not supported rigorously by the data in hand. Indeed, different fusions have different activity levels and off-target effects -- but it is known that different expression levels may have such outcomes and this possibility has not been removed. For example, the increases seen in off-target RNA editing effects with some constructs may directly indicate misfolding and degradation of these proteins leading to increased release of free deaminase. I would suggest at least one demonstration of protein concentration (e.g. westerns tagging both N & C or the whole protein) across different fusions and intercolations in order to validate that changes in activity, especially off-target reductions are indeed due to the steric arrangement of the molecule and not just lower expression levels or degradation.

Minor Comments

Results Section 1:

Propose renaming for clarity (e.g. Activity comparison of Intra-domain and Circularly permuted SpCas9-CBE Fusions)

The authors have developed and tested an impressive number of constructs in this paper however the naming and descriptive scheme is very difficult to follow. To remedy this, do not use Figure 1a to outline every construct tested in the paper but rather outline the type of constructs tested directly prior to demonstrating the data. Moreover, given the naming used in the cartoons is not consistent with what is used later in the data figures it does not help to clarify the following tests. I would suggest either 1. A name simplification or 2. A code be added for each cartoon and construct that is used to maintain name consistency and clarity throughout.

Results Section "Domain engineering of a minimal ABE fine-tunes base editing activity based on protein secondary structure":

Paragraph 2: If you would like to make a claim about domain insertions into the Rec lobe I suggest you try more than one spot in the rec lobe. Otherwise please rephrase: "Insertion within the rec lobe... significantly impeded activity" -- we do not know this to be true of the whole rec lobe (which is significantly larger than the tested area).

Results Section "Intradomain insertion can attenuate the incidence of aberrant off-target RNA editing" Paragraph 2: "Here, we suspect that the altered positioning of the deaminating catalytic pocket is "hidden" from the nucleoplasm, until further R-loop interaction upon binding to its cognate target spacer" please rephrase this conjecture such that it is supported by the data or provide more data to support these "suspicions".

S5 & 12 figures are too small to read easily

Results section "Domain inlaid ABEs enables correction of disease-specific loci and single vector

AAV-media”:

Please provide the % corrected (base edited) for the correct site when discussing your application also the total correct without bystander mutations (i.e. other bases mutated that will change coding) rather than “Proportion of A to G...”. Figures 3 & S14 are tough to decipher. Is this a therapeutic level of editing? If so, this has validated your research. If not, I believe that is OK as you have pushed the molecules closer, but please explain as such.

Figure S14a, please indicate the mutant residue

Methods:

Cycle number for amplicon sequencing is not provided

REVIEWER COMMENTS

Reviewer #1 (Remarks to the Author):

In this article, the authors engineer a set of base editing proteins by fusing CBE/ABE domains to circularly permuted (CP) or inserting CBE/ABE domains (ID) inside SpCas9/SaCas9. The on-target and off-target editing profiles of these Cas9 base editors are evaluated using DNA and RNA sequencing analysis. The authors found that intradomain insertion of ABE in SaCas9 at position I744 improved the DNA on-target editing efficiency and reduced RNA off-target editing compared to N-terminal fusion of ABE to SaCas9. In general, the authors provide an interesting method to increase on-target but reduce promiscuous RNA off-target base editing efficiency. However, the manuscript is not well written and the statements are not convincing with current data. The annotation of engineered Cas9 is confusing and not consistent in the main text, figure legends and tables. A concise nomenclature of Cas9 derivatives is suggested. Many labels used in the figures are not explained in the figure legend and the main text. The quality of figures need to be improved to make results more explicit to readers.

ACTION: The following changes were made such that now all nomenclature reflects only those mentioned in the Figures (and written text has been adapted as such):

→ TAM-AIDx, referring to AIDx (nCas9-AID (P182X)) is now referred to simply as nSpCas9-hAIDx (P182X).

The construct formerly referred to as TAM which describes dCas9 not nCas9 (*Ma et al. 2016, Nature Methods; doi:10.1038/nmeth.4027*)

→ “N-terminal ABEmax” (Figure 1d; also referred to as “ABEmax” [Figure 1c]; also “N-terminal linked SpCas9 ABEmax” [Figure 1a]) is simplified to ABEmax. This naming refers to the widely accepted consensus that ABEmax refers to only N-terminal linked ABE7.9, codon optimized, and placed at the N-terminal of SpCas9. This is not to be confused with ABEmax linked to SaCas9, which is referred to as “SaABEmax” (*Huang et al. 2019, Nature Biotechnology; doi: 10.1038/s41587-019-0134-y*).

→ “ID1058 miniABEmax” (Figure 1c; “ID1058 miniABEmax(V82G)” from Figure 1d) will now be referred to only as ID1058 miniABEmax (V82G). There is no need to refer to it as SpCas9 or SaCas9 as the original publication (*Grünwald et al. 2019, Nature Biotechnology; https://doi.org/10.1038/s41587-019-0236-6*) presumes that “miniABEmax (V82G)” refers to only the SpCas9 variant, of which no SaCas9 variant has been hitherto generated.

→ “CP1029 C miniABEmax”, “CP1029 N/C miniABEmax”, “CP1029 N miniABEmax” (Figure 1a&c) will be referred to only as “CP1029 C-terminal miniABEmax (V82G)”, “CP1029 N-terminal miniABEmax (V82G)”, “CP1029 N/C-terminal miniABEmax (V82G)”. The specification of N or C or N/C split terminal was necessary as these constructs have not been defined yet (in a circularized format) in the literature.

→ “ID1058 ABEmax” will remain as it presently stands.

→ “SaCas9-ABEmax” (Figure 2B; also referred to as “N-terminal ABEmax” [Figure 2c-d]), is simplified to SaABEmax (in line with the formal name given from previous publication, *Richter et al. 2020 Nature Biotechnology; https://doi.org/10.1038/s41587-020-0453-z*)

→ “SaCas9-miniABEmax(V82G)” (Figure 2B, also referred to as “N-terminal miniABEmax (V82G)” [Figure 2c-d]) is simplified to Sa-miniABEmax (V82G), and a statement defining that Sa-miniABEmax(V82G) refers to N-terminal-linked miniABEmax (V82G) appended to SaCas9 is included. This nomenclature is in line with the most recent publication of this variant (*Grünwald et al. 2020 Nature Biotechnology; https://doi.org/10.1038/s41587-020-0535-y*)

→ “microABE I744” (also “Sa-I744” [Figure 2b], “ID744 miniABEmax(V82G)” [Figure 2c], “ID744 miniABEmax (V82G)” [Figure 2d]) will now only be referred to as “microABE I744”

→ Intradomain-SaCas9n N730, and Intradomain-SaCas9n G129 referred to as “ID129 miniABEmax(V82G)” and “ID730 miniABEmax(V82G)” will be referred to as “Sa-ID129 miniABEmax (V82G)” and “Sa-ID730 miniABEmax (V82G)”, respectively, to better reflect the naming convention of SaABEmax, Sa-miniABEmax (V82G), in which Sa and the term ID (intradomain) is placed.

Comments:

1. The schematic in Fig. 1a lacks of detailed information on how each construct is made. The DNA construct map (similar to Fig. 3a) should be provided. The nomenclature of the construct is confusing and inconsistent.

ACTION: Figure 1a has now been removed, and in part updated to reflect hopefully a more standard /better naming nomenclature. A schematic ideogram of the construct (similar to our final AAV-DNA construct map) has now been inserted. We hope that the simplified layout will reduce potential confusion.

2. The editing window for different engineered SpCas9 may be different. In Fig. 1b, only cytosine-to-guanine transversion efficiency at position 9 of the sgRNA is presented. The editing efficiency of other positions are also needed to compare the editing efficiency. The Fig. 1b is suggested to move to the supplementary file as this result is not related to the core topic (Cas9 ABE). The results of this figure are not clearly described in the main text on Page 5.

ACTION: We thank the reviewer for their comment, and agree that other cytosine positions within that particular *YFP*-targeting sgRNA and window needed to be more clearly considered. To address this we have moved Fig. 1b to the Supplementary section. While attempting to “zero” in on the ideal construct which would form the basis for a deeper exploration of intradomain and circularly permuted engineering, we considered the cytosine nucleotide in the protospacer that was most highly edited (position 9) for nSpCas9-hAIDx (formerly TAM-AIDx); in Supplementary Figure 2b, the full activity window is shown in relation to other cytosine bases as a nucleotide quilt showing the comparison with the other intradomain and circularly permuted constructs. We have moved this nucleotide quilt to replace Fig. 1b, though, as you can see, position 9 (as previously shown) was the chief site of editing. In Supplementary Figure 4, a more detailed comparison against other CBEs is shown, which also details the activity window and effects on permutating the base editor.

The statement: “Collectively, we show that the intradomain insertion of the hAIDx deaminase maintains a consistent on-target DNA signature (characterized by cytosine-to-guanine transversions at position 9 of the sgRNA) compared to its C-terminal variant, and that nSpCas9 domain-interruptions are most amenable at residue 1058 in our preliminary screen (**Fig. 1b; Supplementary Fig. 2**)” was inserted to more clearly detail the final choice for intradomain (position 1058) and circularly permuted (position 1029) engineering. With regards to the choice to show “cytosine-to-guanine” transversion as opposed to “cytosine-to-thymine” conversion, we felt that using the cytosine-to-guanine transversion activity of nSpCas9-hAIDx (formerly TAM-AIDx) was the most reliable readout for on-target activity given that C>G transversions accurately and representatively reflected the degree of enzyme activity when compared to other transition or transversions catalyzed by nSpCas9-hAIDx (formerly TAM-AIDx). Initially, a statement regarding product purity was intended, though this lacked statistical support. Nonetheless, it is interesting that the behaviour of the hAIDx deaminase is unchanged by these permutations, and hence, we sought to emphasize that with a focus on the C>G transversion.

In summary, **Figure 1b** was moved to the Supplementary section, whereas what was **Supplementary Figure 2a** was moved to the main text. This is to highlight the steps involved in the optimization and final choice of residue 1058 and residue 1029 for intradomain and circularly permuted constructs as these two positions formed the basis for subsequent comparative analyses.

3. In Fig. 1c, the nomenclature is quite confusing and it is hard to read.

ACTION: As outlined above, the nomenclature of each construct now follows a standardized format and naming that will hopefully make it easier for readers to differentiate between the different constructs.

4. In Fig. 1d, 2c and Supplementary Fig. 6, the calculation of the off-target RNA editing (A-to-I) activity should be provided in figure legend.

ACTION: Further explanations as to the calculation and scaling (in R) is provided in the methods section under 'Amplicon sequencing analysis': "Heatmaps quantifying the off-target profiles were generated in R (v. 1.2.5019) using the 'superheat' package."

Additionally, the figure legend for Supplementary Fig. 6 has been updated to include the following sentence: "Results display the average adenosine to inosine editing at each adenosine nucleotide position within the amplicon and scaled accordingly using the 'superheat' package in R (v. 1.2.5019)."

5. Fig. 2a is suggested to move to Supplementary Information.

ACTION: We have now moved Figure 2a to the Supplementary Figure Section.

6. In Fig. 2b, how to calculate the composite A-to-G efficiency should be described in the figure legend. Meanwhile, the data for each sgRNA targeting locus should be included in the figure or in Supplementary Information.

ACTION: To avoid any confusion we have moved Supplementary Fig. 11 to now be included as main Fig. 3a. As such, Fig. 3 now displays the individual activity profiles at each sgRNA target locus, used to determine the final activity window, which is now in panel b of Fig 3. This is in keeping with recent publications (such as *Zhang et al. 2020, Nature Biotechnology*; <https://doi.org/10.1038/s41587-020-0527-y> or *Grünwald et al. 2020, Nature Biotechnology*; <https://doi.org/10.1038/s41587-020-0535-y>), which shows all data points as part of the main figure as well as a composite of the data points for easy readability.

7. In Fig. 2d, the distribution of off-target editing in chromosomes seems unnecessary information. The total number of off-target editing occurrences should be provided.

ACTION: The total number of edits has now been inserted into the text and emphasized more clearly as a discussion point: "The microABE I744 dramatically lowered the incidence of aberrant mRNA off-target events compared to both the SaABEmax and Sa-miniABEmax (V82G) (2243 reads containing adenosine-to-inosine editing for microABE I744 as compared to 4425 and 52,030 reads for Sa-miniABEmax [V82G] and Sa-ABEmax, respectively)."

Fig. 2d has been changed to the Supplementary section (Supplementary Figure 15) and is discussed in the context of newly added whole-exome sequencing data, in which the gDNA from samples used for RNA off-target profiling in Fig. 2d was sequenced as part of our off-target analyses.

8. In Fig. 3, the positive control (e.g. SaCas9-ABEmax in one or two AAV vector(s)) is missing. The scale of proportion of A to G conversion is different from what are used in other figures, which makes the results hard to compare with each other. The description of semi-rigid linker is missing.

ACTION: Thank you for the comment. We were interested in analysing the packagability of our construct as an all-in-one single vector ABE, and its capacity to effect A-to-G conversions in this format. As such, the use of SaABEmax would require dual-delivery of both an AAV-packaged SaABEmax, as well as its sgRNA. We have added a caveat in the discussion that we did not compare its activity against SaABEmax packaged into dual-vector AAV. This is mentioned as the following sentence in the discussion:

"Given these promising results however, which maximizes upon the packaging constraints of AAV-delivered payloads, future directions will consider further optimization to the payload architecture by placing the U6-sgRNA component in the antisense direction, or adding additional regulatory elements for enhanced protein expression, as well as comparing our single-vector format against dual-vector constructs such as packaged SaABEmax.²³"

Although we recognize the value of performing such a head-to-head comparison; for the time being, we have reserved this topic as subject to ongoing and future research in which we further optimize the deliverability of the microABE in preparation for *in vivo* work. With regards to the editing scale for the proportion of A-to-G conversions, recent publications (*Levy et al. 2020, Nature Biomedical Engineering*;

<https://doi.org/10.1038/s41551-019-0501-5>) suggested that further optimization is needed to achieve a more obvious editing effect. At the time these experiments were performed, the manuscript submitted by Levy and colleagues was not yet published, and this information was unavailable to us. For example, we are continuing efforts to further optimize the orientation of the U6-sgRNA component, as well as exploring the use of other terminator sequences, such as the truncated WPRE element highlighted by Levy and colleagues. Of course, these alterations will require the use of dual-vector AAV systems as the addition of other regulatory elements such as the truncated WPRE adds an additional 250 bp, which would require extensive profiling as this would mean that both single-vector and dual-vector characterization and architecture optimization is required. Therefore, the head-to-head comparison with SaABEmax would be more apt for this area of study than in this current research. At the moment however, we have added an extra cell line by targeting H9-derived retinal optic cup organoids to provide more AAV data.

In addition, we have undertaken novel methods to deep sequence the AAV virus containing our packaged microABE I744 to show that truncation of the AAV has not occurred as a result of our all-in-one packaging construct when used in tandem with the SCP1 promoter. Interestingly, genomic rearrangement events were also few as shown in **Figure 4a-b**. We have undertaken this work given that we were at the very limit of the AAV-packaging limit, and were hoping to pinpoint why we did not achieve more obvious editing. We postulated that there may have been truncation at the 3' end of the AAV construct at the right ITR, which is where the sgRNA lies. However, our adapted sequencing platform showed that this has not occurred. We would like to note that this is the first instance in which such sequencing has been performed as a means of quality-control for AAV-packaged base editors, and may be valuable as a technical point-of-interest for work based on *in vivo* AAV-base editing. From this, we inserted additional Discussion to explain that low editing efficiency is most likely due to the arrangement, architecture and non-coding features of the AAV vector, and that insofar, it does not appear to be due to the truncation of the AAV vector in this particular instance. Levy and colleagues identified that the sgRNA placed near an ITR can negatively impact on-target editing efficiencies. Therefore, further ongoing work to optimize the AAV vector also includes the reversion of the U6-sgRNA portion of the AAV vector so that it is placed on the anti-sense strand. Please note however, that due to the incapacitating nature of COVID-19 in our laboratory, it is rather difficult at the time being to extensively pursue these topics in this particular project.

We have provided a description of the semi-rigid linker in the Results section (including the amino acids that make up the semi-rigid linker), as well as updated the figure legend to reflect this.

9. In Supplementary Fig. 5, no statistical data are provided. Therefore, the “significantly” used in the statement on page 5 is inappropriate.

ACTION: We have removed the term “significantly” from the statement referring to Supplementary Figure 5.

Reviewer #2 (Remarks to the Author):

In this paper Tran et al. design and test a large number of interesting base editor fusions in HEK293 cells based on extrapolations from extensive prior work done in the CRISPR field. Nevertheless their construct design is astute and by combining and directly comparing a number of past observations they validate much of this previous work and help to bolster previous hypotheses concerning the modular and conserved nature of domain insertion. Moreover they design a specific ABE construct that may be of use in AAV based delivery in the future.

I find this work to be both useful and of high interest and recommend publication with few modifications.

Major Issues:

In order to better demonstrate the importance of this work please include an experiment demonstrating the activity in at least one, preferably multiple, cell types other than HEK293s. Preferably, a primary cell type. Any animal model, reporter or otherwise with AAV would be ideal but animal demonstration should not be required for publishing

ACTION: We thank the reviewer for their comment and have now included additional data showing editing with AAV-packaged microABE 1744 against iPSC-derived retinal optic cup organoids (H9). We selected the use of H9 optic cup organoids as we wanted to select a terminally differentiated cell line as other publications in the field have identified the ongoing division of immortalized cell lines as being a significant impediment to reproducible AAV-base editing (*Levy et al. 2020, Nature Biomedical Engineering; <https://doi.org/10.1038/s41551-019-0501-5>*). Please find the data on this in **Supplementary Figure 17**. In addition, we have also included deep sequencing data on the AAV-packaged virus containing the microABE 1744 to show that 5' or 3' truncation has not occurred as a result of the single-vector format, and that genomic rearrangement events were few. It should be noted that this method of sequencing has not been previously demonstrated against AAV-packaged base editors and may represent a technical point-of-interest that could benefit the quality-control process for future *in vivo* base editing. As part of our ongoing work, recent publications have identified that further optimization to the architecture of the AAV construct is needed in order to fully realize higher editing efficiencies. Currently, we continue further work on the optimization of our AAV-packaged base editor, which includes features such as placing the U6-sgRNA component on the anti-sense strand, as well as exploration of the use of other promoters to further drive expression and the addition of further regulatory elements (Please see response to Reviewer #1). Levy and colleagues (*Levy et al. 2020, Nature Biomedical Engineering; <https://doi.org/10.1038/s41551-019-0501-5>*) has demonstrated the importance of such optimizations, and we are urgently pursuing further work at the moment to explore the effects on on-target efficiency that such an alteration has on the AAV-deliverability of our construct. Please note however, that due to the incapacitating nature of COVID-19 on our laboratory, it is rather difficult at the time being to extensively pursue these topics in this particular project.

A number of times throughout the paper comments are made about "stereochemistry" dictating the molecular activity of fusions and intercolations. The use of the term stereochemistry is not only confusing, it is misleading as most of the molecules tested are not the same in composition and therefore can not be construed to be stereoisomers. Please rephrase.

ACTION: We have now removed all references to stereochemistry in the text and refer specifically to the alteration of the base editor arrangement with respect to either SpCas9 or SaCas9 to avoid any confusion.

The assertion that the difference of effects observed is due solely to the steric limitations of specific fusions, while likely, is not supported rigorously by the data in hand. Indeed, different fusions have different activity levels and off-target effects -- but it is known that different expression levels may have such outcomes and this possibility has not been removed. For example, the increases seen in off-target RNA editing effects with some constructs may directly indicate misfolding and degradation of these proteins leading to increased

release of free deaminase. I would suggest at least one demonstration of protein concentration (e.g. westerns tagging both N & C or the whole protein) across different fusions and intercolations in order to validate that changes in activity, especially off-target reductions are indeed due to the steric arrangement of the molecule and not just lower expression levels or degradation.

ACTION: We have firstly revised our claims and suggested that the altered RNA and DNA editing profile that we observe may be due to not only the rearrangement of the secondary structure of the protein, but also due to the incidence of premature STOP codons affecting the expression and activity of the protein, and how this relates to the position of the deaminase domain. Further, we have revised our language to reflect that the changes may be due to misfolding and/or degradation of the base editor, and that the rhythm of protein translation (despite the same codon usage and nucleotide sequence) could also affect the mutagenic profile.

Furthermore, we have performed western blots on the key constructs from which these claims were based. Included as a main figure (**Figure 3d**), western blots using a primary antibody targeting the N terminus and the C terminus were performed on constructs Sa-ID129 miniABEmax (V82G), Sa-ID730 miniABEmax (V82G), microABE I744, SaABEmax, and Sa-miniABEmax (V82G). Unfortunately, we were unable to perform a western blot on the intercalating linker due to the lack of an existing primary antibody. Nonetheless, as shown by **Figure 3d**, the banding size of each construct firmly reflects the expression of the expected protein for both the N and the C terminus. We feel that this strongly supports our point that the effects on RNA off target effects is mostly due to the specific arrangement of the ABE, as opposed to other effects such as the incidence of premature stop codons releasing the deaminase domain. We cannot however preclude this as being the cause for our observations, and so we have firstly removed all instances of the word "stereochemistry" and "steric arrangement" added the following points in the Discussion section: "Here, we reasoned that the inlaying of a base editor domain could further attenuate the incidence of RNA-off target events by exerting either a steric limitation on the deaminase domain, or by altering the secondary structural folding and expression of the base editor."

Minor Comments

Results Section 1:

Propose renaming for clarity (e.g. Activity comparison of Intra-domain and Circularly permuted SpCas9-CBE Fusions)

ACTION: As suggested we have now changed the subheading from: "Probing the effects of protein secondary structural rearrangements of SpCas9-CBEs" to "Activity comparison of Intra-domain and Circularly permuted SpCas9-CBE Fusions"

The authors have developed and tested an impressive number of constructs in this paper however the naming and descriptive scheme is very difficult to follow. To remedy this, do not use Figure 1a to outline every construct tested in the paper but rather outline the type of constructs tested directly prior to demonstrating the data. Moreover, given the naming used in the cartoons is not consistent with what is used later in the data figures it does not help to clarify the following tests. I would suggest either 1. A name simplification or 2. A code be added for each cartoon and construct that is used to maintain name consistency and clarity throughout.

ACTION: We acknowledge the confusion regarding the complex construct names and have taken the steps to ensure that a standardized naming system is used for each construct that should be more intuitive and in line with the current literature. Secondly, Figure 1a was moved to the Supplementary Figure, and was subsequently updated and simplified. We have decided that, prior to each head-to-head comparison, that a smaller subfigure be added to show a DNA construct map of the constructs being compared as an overview. This work is divided into two different comparisons; work comparing SpCas9-based constructs, and work comparing SaCas9-based constructs and we agree entirely with your assessment that this delineation needs to be broken up.

The following changes were made; all nomenclature now reflects only those mentioned in the Figures (and written text has been adapted as such):

→ TAM-AIDx, referring to AIDx (nCas9-AID (P182X)) is now referred to simply as nSpCas9-hAIDx (P182X).

The construct formerly referred to as TAM which describes dCas9 not nCas9 (*Ma et al. 2016, Nature Methods; doi:10.1038/nmeth.4027*)

→ “N-terminal ABEmax” (Figure 1d; also referred to as “ABEmax” [Figure 1c]; also “N-terminal linked SpCas9 ABEmax” [Figure 1a]) is simplified to ABEmax. This naming refers to the widely accepted consensus that ABEmax refers to only N-terminal linked ABE7.9, codon optimized, and placed at the N-terminal of SpCas9. This is not to be confused with ABEmax linked to SaCas9, which is referred to as “SaABEmax” (*Huang et al. 2019, Nature Biotechnology; doi: 10.1038/s41587-019-0134-y*).

→ “ID1058 miniABEmax” (Figure 1c; “ID1058 miniABEmax(V82G)” from Figure 1d) will now be referred to only as ID1058 miniABEmax (V82G). There is no need to refer to it as SpCas9 or SaCas9 as the original publication (*Grünwald et al. 2019, Nature Biotechnology; <https://doi.org/10.1038/s41587-019-0236-6>*) presumes that “miniABEmax (V82G)” refers to only the SpCas9 variant, of which no SaCas9 variant has been hitherto generated.

→ “CP1029 C miniABEmax”, “CP1029 N/C miniABEmax”, “CP1029 N miniABEmax” (Figure 1a&c) will be referred to only as “CP1029 C-terminal miniABEmax (V82G)”, “CP1029 N-terminal miniABEmax (V82G)”, “CP1029 N/C-terminal miniABEmax (V82G)”. The specification of N or C or N/C split terminal was necessary as these constructs have not been defined yet (in a circularized format) in the literature.

→ “ID1058 ABEmax” will remain as it presently stands.

→ “SaCas9-ABEmax” (Figure 2B; also referred to as “N-terminal ABEmax” [Figure 2c-d]), is simplified to SaABEmax (in line with the formal name given from previous publication, *Richter et al. 2020 Nature Biotechnology; <https://doi.org/10.1038/s41587-020-0453-z>*)

→ “SaCas9-miniABEmax(V82G)” (Figure 2B, also referred to as “N-terminal miniABEmax (V82G)” [Figure 2c-d]) is simplified to Sa-miniABEmax (V82G), and a statement defining that Sa-miniABEmax(V82G) refers to N-terminal-linked miniABEmax (V82G) appended to SaCas9 is included. This nomenclature is in line with the most recent publication of this variant (*Grünwald et al. 2020 Nature Biotechnology; <https://doi.org/10.1038/s41587-020-0535-y>*)

→ “microABE I744” (also “Sa-I744” [Figure 2b], “ID744 miniABEmax(V82G)” [Figure 2c], “ID744 miniABEmax (V82G)” [Figure 2d]) will now only be referred to as “microABE I744”

→ Intradomain-SaCas9n N730, and Intradomain-SaCas9n G129 referred to as “ID129 miniABEmax(V82G)” and “ID730 miniABEmax(V82G)” will be referred to as “Sa-ID129 miniABEmax (V82G)” and “Sa-ID730 miniABEmax (V82G)”, respectively, to better reflect the naming convention of SaABEmax, Sa-miniABEmax (V82G), in which Sa and the term ID (intradomain) is placed.

Results Section “Domain engineering of a minimal ABE fine-tunes base editing activity based on protein secondary structure”:

Paragraph 2: If you would like to make a claim about domain insertions into the Rec lobe I suggest you try more than one spot in the rec lobe. Otherwise please rephrase: “Insertion within the rec lobe... significantly impeded activity” -- we do not know this to be true of the whole rec lobe (which is significantly larger than the tested area).

ACTION: We have rephrased the wording to omit reference to the REC lobe. The insertion at this position was meant to be representational of that area based on some previous data generated by Oakes and colleagues (*Oakes et al. 2016, Nature Biotechnology; DOI: 10.1038/nbt.3528*). However, we recognize that it may not be feasible to make the definitive claim about the REC lobe without some crystal structure data, or much more extensive profiling of this area, and as such we have revised the wording and the claims.

Phrasing of the paragraph has now been changed to the following: “Interestingly, the insertion of a base editing domain between residues 119 to 132 in SaCas9n significantly impeded the on-target activity of the miniABEmax (V82G) (between 0.00 and 5.47% across residues 119 to 132), whereas on-target activity

was dramatically improved when inserted between residues 730 to 745 of SaCas9n (between 5.39 and 17.7% across residues 730 to 745).”

Results Section “Intradomain insertion can attenuate the incidence of aberrant off-target RNA editing”
Paragraph 2: “Here, we suspect that the altered positioning of the deaminating catalytic pocket is “hidden” from the nucleoplasm, until further R-loop interaction upon binding to its cognate target spacer” please rephrase this conjecture such that it is supported by the data or provide more data to support these “suspicions”.

ACTION: We agree that without supporting crystallography data, we cannot make such a claim and therefore, retract this conjecture. We have rephrased this to now read: “Here, we postulate that the altered positioning of the deaminating catalytic pocket is “hidden” from the circulating RNA transcripts, though we cannot definitively preclude other mechanisms that would affect the RNA mutagenicity of domain inlaid ABEs without crystallographic structures.”

Additionally, we have included the following sentences: “Next, we wanted to determine if whether these differences in RNA-off target effects were due to variations in protein expression or to ectopic protein misfolding.¹⁰ We performed western blots with primary antibodies targeting the N-terminus of domain-inlaid nSaCas9 base editors and the C-terminal flag tag of the respective constructs. Overall, there was no major difference in protein expression and expected banding patterns for each construct. Taken together, these results indicate that it was unlikely that RNA off-target-specific differences were attributable to protein-expression and folding specific variations, such as premature stop codons occurring within the open reading frame (**Fig. 3d; Supplementary Fig. 14**).”

S5 & 12 figures are too small to read easily

ACTION: The text in these figures has been enlarged.

Results section “Domain inlaid ABEs enables correction of disease-specific loci and single vector AAV-media”:

Please provide the % corrected (base edited) for the correct site when discussing your application also the total correct without bystander mutations (i.e. other bases mutated that will change coding) rather than “Proportion of A to G...”. Figures 3 & S14 are tough to decipher. Is this a therapeutic level of editing? If so, this has validated your research. If not, I believe that is OK as you have pushed the molecules closer, but please explain as such.

ACTION: We have now mentioned that it is unclear in the discussion whether this constitutes a therapeutically relevant level of editing. Additionally, please see response to Reviewer #1 regarding the editing efficiency, as well as the response detailed above regarding future directions. Upon contacting Levy and colleagues (*Levy et al. 2020, Nature Biomedical Engineering*; <https://doi.org/10.1038/s41551-019-0501-5>) for further details regarding how they performed *in vitro* AAV-base editing, we firmly believe that there are a number of factors that are much beyond the current scope of this work that require optimization. As stated above, this includes optimization to the architecture, as well as possibly different capsid derivatives (such as Anc80). We are currently exploring architectural optimizations, but have since made it clear that further studies are required.

Figure S14a, please indicate the mutant residue

ACTION: The disease-causing variant being targeted is now clearly demarcated, and the corresponding amino acid sequence has also been inserted.

Methods:

Cycle number for amplicon sequencing is not provided

ACTION: We have amended the methods to indicate that a V3 MiSeq flow cell was used with a 2 x 251 bp cycle.

Reviewers' Comments:

Reviewer #1:

Remarks to the Author:

In general, the quality of the manuscript has been improved. However, I have a few minor concerns.

1. In Figure 4c and Supplementary Figure 16a, how the "proportion of A to G conversion x 1000" was calculated? The standard percentage calculation should be used in raw data processing for comparison with the data in Supplementary Figure 7a, 7b, 9 and 10. It seems that the editing efficiency of AAV delivered microABE I744 is quite low, which makes the results less interesting. Instead of using the vague description (like "modest editing" and "observable editing") in the Result Section, authors are suggested to describe the explicit editing efficiency.

2. Similarly, the calculation of "off-target RNA-editing (A-to-I) activity" should be described in the figure legend or methods.

3. The Supplementary Figure 5 and 12 are flipped.

Reviewer #2:

Remarks to the Author:

I thank the authors for their revisions of this manuscript the figures and text are clearer now. With minor changes I recommend publication

Minor comments:

The new westerns should be quantified via densitometry or rerun with more equal loading before any conclusion can be drawn from them (re: line 265); it is very hard to draw anything from gels that have been unequally loaded to such an extent based on the loading control.

The authors claim modest editing in figure 4c and editing in figure S16a & S17 but they change the scale from other figures previous to this. The previous "percentage [C to T/A to G] conversion" is a more straightforward measurement to understand than "Proportion A to G conversion x1000" Please change all of the latter into the former. If this was done because the editing percent is low, please call the low percentages out and explain that the constructs need optimization but that there is activity, do not hide it in a different scale.

The authors claim "Nonetheless, it remains to be determined if these editing efficiencies are at a therapeutically relevant threshold." This is something that can be ascertained based on past data on previous gene therapy in the eye and the spread of AAV (i.e >10% of PR in the retina). If you are not at this threshold, which I do not think is the case, please just say so.

Remove "significant" and "forward" on line 279 unless the comment above is addressed. While the concept presented here is intriguing and more elegant than the dual vector approach with only one data point not in HEK cells, and minimal editing in this scenario, we cannot be sure that this represents a more useful tool for clinical translation than the dual constructs previously published. While outside the scope of this work the authors would have to run head to head in vivo studies and look at off target effects with the published dual constructs to demonstrate this. I would strongly recommend this as the next step in this line of research.

REVIEWER COMMENTS

Reviewer #1 (Remarks to the Author):

In general, the quality of the manuscript has been improved. However, I have a few minor concerns.

1. In Figure 4c and Supplementary Figure 16a, how the “proportion of A to G conversion x 1000” was calculated? The standard percentage calculation should be used in raw data processing for comparison with the data in Supplementary Figure 7a, 7b, 9 and 10. It seems that the editing efficiency of AAV delivered microABE I744 is quite low, which makes the results less interesting. Instead of using the vague description (like “modest editing” and “observable editing”) in the Result Section, authors are suggested to describe the explicit editing efficiency.

Action: To avoid any confusion we now report this as a percentage, rather than proportion. All figures and corresponding captions have been updated accordingly. We have also directly stated the editing efficiency and removed the terms “observable” and “modest”.

2. Similarly, the calculation of “off-target RNA-editing (A-to-I) activity” should be described in the figure legend or methods.

Action: In the Methods Section - Amplicon sequencing analysis: we have now specifically outlined how the off-target A-to-I activity was calculated.

3. The Supplementary Figure 5 and 12 are flipped.

Action: These supplementary figures have now been rotated.

Reviewer #2 (Remarks to the Author):

I thank the authors for their revisions of this manuscript the figures and text are clearer now. With minor changes I recommend publication

Minor comments:

The new westerns should be quantified via densitometry or rerun with more equal loading before any conclusion can be drawn from them(re: line 265); it is very hard to draw anything from gels that have been unequally loaded to such an extent based on the loading control.

Action: We have now quantified the western blot via densitometry, and have inserted this information in the accompanying supplementary figure (now Supplementary Figure 16). In addition, we have tempered the claims regarding “protein expression,” removing the term from the following sentence: “Taken together, these results indicate that it was unlikely that RNA off-target-specific differences were attributable to ~~protein-expression~~ and folding specific variations, such as premature stop codons occurring within the open reading frame”

The authors claim modest editing in figure 4c and editing in figure S16a & S17 but they change the scale from other figures previous to this. The previous “percentage [C to T/A to G] conversion” is a more straightforward measurement to understand than “Proportion A to G conversion x1000” Please change all of the latter into the former. If this was done because the editing percent is low, please call the low percentages out and explain that the constructs need optimization but that there is activity, do not hide it in a different scale.

Action: As outlined above, we now report this as a percentage, rather than proportion. All figures and corresponding captions have been updated accordingly.

The authors claim “Nonetheless, it remains to be determined if these editing efficiencies are at a therapeutically relevant threshold.” This is something that can be ascertained based on past data on previous gene therapy in the eye and the spread of AAV (i.e >10% of PR in the retina). If you are not at this threshold, which I do not think is the case, please just say so.

Action: This section of the Discussion has now been re-written to now read: “Currently, it is unlikely that the editing efficiency of our single AAV vector-packaged microABE I744 has surpassed a therapeutic threshold. Nonetheless, given our promising *in vitro* plasmid-based results, future directions could consider further optimization to the AAV payload architecture by placing the U6-sgRNA component in the antisense direction, or adding additional regulatory elements for enhanced protein expression, as well as comparing our single-vector format against dual-vector constructs such as packaged SaABEmax or through the screening different promoter sequences.²³”

Remove “significant” and “forward” on line 279 unless the comment above is addressed. While the concept presented here is intriguing and more elegant than the dual vector approach with only one data point not in HEK cells, and minimal editing in this scenario, we cannot be sure that this represents a more useful tool for clinical translation than the dual constructs previously published. While outside the scope of this work the authors would have to run head to head *in vivo* studies and look at off target effects with the published dual constructs to demonstrate this. I would strongly recommend this as the next step in this line of research.

Action: We have removed the sentence, which was on line 279 and contained the words “significant” and “forward”. We are also grateful for the insight and suggested experiments.